# Impact of Tumor Size on Prognosis in Differentiated Thyroid Cancer with Gross Extrathyroidal Extension to Strap Muscles: Redefining T3b

**DOI:** 10.3390/cancers16142577

**Published:** 2024-07-18

**Authors:** Joonseon Park, Solji An, Ja Seong Bae, Kwangsoon Kim, Jeong Soo Kim

**Affiliations:** Department of Surgery, College of Medicine, The Catholic University of Korea, Seoul 06591, Republic of Korea; joonsunny@naver.com (J.P.); solji1130@hanmail.net (S.A.); noar99@naver.com (K.K.); btskim@catholic.ac.kr (J.S.K.)

**Keywords:** differentiated thyroid cancer, tumor size, gross extrathyroidal extension, T3b, staging, survival analysis, AJCC/UICC TNM stage

## Abstract

**Simple Summary:**

This study investigated the impact of tumor size on T3b differentiated thyroid cancer prognosis. No significant difference was found in the prognosis of small T3b tumors compared to the T1 tumors. Disease-specific survival, disease-free survival, and overall survival were significantly lower only in large T3b tumors compared to T2 and T3a. If T3b tumors are 2 cm or smaller, downstaging may be considered. The modified T category, reclassifying T3b (≤2 cm) as T1, showed better staging performance than the existing category. Adopting this modified T category could improve the prognostic accuracy of the AJCC/TNM staging.

**Abstract:**

The prognostic significance of tumor size in T3b differentiated thyroid cancer (DTC) remains debated and underexplored. This study aimed to examine the varying impact of T3b based on tumor size, analyzing disease-specific survival, disease-free survival, and overall survival. A retrospective review of 6282 DTC patients who underwent thyroid surgery at Seoul St. Mary’s Hospital from September 2000 to December 2017 was conducted. T3b was classified into three subcategories, T3b-1 (≤2 cm), T3b-2 (2–4 cm), and T3b-3 (>4 cm), using the same size criteria for T1, T2, and T3a. T3b-1 showed no significant difference in disease specific survival compared to T1, and both disease-free and disease-specific survival curves were sequentially ranked as T1, T3b-1, T2, T3a, T3b-2, and T3b-3. The modified T category, reclassifying T3b-1 as T1, demonstrated superior staging performance compared to the classic T category (c-index: 0.8961 vs. 0.8959 and AUC: 0.8573 vs. 0.8518). Tumors measuring 2 cm or less within the T3b category may require downstaging, and a modified T category could improve the precision of prognostic staging compared to the current T category.

## 1. Introduction

Differentiated thyroid cancer (DTC) is widely recognized for its favorable prognosis due to its low disease-specific mortality (DSM) rate [1,2,3,4,5]. Despite the generally low DSM of DTC, the American Joint Committee on Cancer/Union for International Cancer Control (AJCC/UICC) TNM staging system determines each tumor (T), regional lymph node (N), and distant metastasis (M) category based on their impact on disease-specific survival (DSS) [6,7,8]. Therefore, the AJCC/UICC TNM staging system should clearly indicate the stratification of DSS corresponding to each disease stage. The current eighth edition of the AJCC/TNM staging system primarily differentiates between T1, T2, and T3a based on tumor size [9,10,11]. However, the classifications of T3b, T4a, and T4b are determined by the presence of cancer invasion into the surrounding structures. Specifically, T3b is defined by gross extrathyroidal extension (gETE) into the strap muscles, determined by the surgeon’s visual assessment, irrespective of the tumor size [9,12,13,14,15].

In 2017, the redefinition of T3b was based on gETE rather than minimal ETE (mETE). In response to this, numerous studies have been conducted to validate the change in the T3b classification [16,17,18,19,20,21,22]. Several studies suggest that gETE limited only to the strap muscles does not affect the prognosis [16,18,19,20,21], while other research indicates that it significantly worsens the prognosis [23,24,25,26]. Such inconsistent results may be due to overlooking the impact of the tumor size in T3b. Several studies have evaluated the influence of T3b based on the size of the primary tumor [19,20,21]. In our prior institutional research, we established that there was no difference in recurrence rates between T3b with a small tumor size and T2 disease [17]. However, the study was constrained by limitations, such as solely investigating disease-free survival (DFS) instead of DSS, and omitting T1 and T3a. Building on these findings, our objective is to elucidate the impact of T3b across all tumor size categories. We conducted a thorough prognosis assessment covering DFS, DSS, and overall survival (OS) across not only T2, but also T1 and T3a categories.

This study aimed to clarify the significance of tumor size in T3b on the DFS and DSS of DTC through comparing T3b subcategories with other T categories and by analyzing the risk factors for DSM. Ultimately, our goal is to propose a modified T category.

## 2. Materials and Methods

### 2.1. Patients

We conducted a retrospective review on 6811 patients with DTC who underwent thyroid surgery at Seoul St. Mary’s Hospital (Seoul, South Korea) from September 2000 to December 2017. The exclusion criteria for this study included 31 patients detected with distant metastasis at the time of their initial diagnosis, 231 patients who underwent initial surgery at other hospitals, 182 patients with incomplete data, and 85 patients lost to follow-up. Ultimately, a total of 6282 patients were incorporated into the study (Figure 1). Clinicopathological information was validated through pathologic reports, with the exception of T3b, which was confirmed intraoperatively by the surgeons [17]. This study was conducted in accordance with the principles stated in the Declaration of Helsinki (revised in 2013). The Institutional Review Board of Seoul St. Mary’s Hospital at the Catholic University of Korea has approved the research protocol (IRB No: KC22RISI0611 and date of approval: 30 August 2022), and due to the retrospective nature of the study, the requirement for informed consent was waived.

### 2.2. Subcategories of T3b Category

To analyze the impact of tumor size on T3b, it was divided based on the size criteria that differentiate the traditional classifications of T1, T2, and T3a. Therefore, T3b was subdivided into T3b-1 (≤2 cm), T3b-2 (2–4 cm), and T3b-3 (>4 cm), and these were included in the T subcategories. (Figure 1).

### 2.3. Perioperative Management and Follow-Up Evaluation

All patients received preoperative evaluation and follow-up in accordance with the 2015 ATA management guidelines [27]. Physical examinations, serum thyroid function tests, thyroglobulin, and anti-thyroglobulin antibody measurements were performed at 2 weeks, 3 months, and 6 months after surgery, and were subsequently conducted annually. Neck ultrasound was conducted annually. Patients who needed additional radioactive iodine (RAI) ablation underwent treatment at least 12 weeks post-thyroidectomy, and whole-body scans were conducted approximately 1 week following the RAI ablation. Patients suspected of recurrence underwent additional imaging procedures, such as computed tomography, positron emission tomography/computed tomography, and RAI whole-body scans, to determine the location and severity of the recurrence. The confirmation of disease recurrence was accomplished through a pathological diagnosis from ultrasound-guided fine-needle aspiration/core needle biopsy or surgical biopsy. The mortality rate data were supplied by the Cancer Center Operations Team at Seoul St. Mary’s Hospital and Central Cancer Registry data, which is based on death records from the Korean Statistical Office.

### 2.4. Primary and Secondary Endpoints

The primary endpoint was a comparison of DSS and DFS among the subclassified T categories, and the secondary endpoint was the predictive accuracy of DSS between the traditional and newly modified T categories.

### 2.5. Statistical Analysis

Continuous variables are represented by means and standard deviations, and numbers described in percentages represent categorical variables. Student’s *t*-test was used to compare continuous variables. To investigate the differences in categorical features among T subcategories, either Pearson’s chi-square test or Fisher’s exact test was used. Univariate and multivariate Cox regression models were utilized to validate significant DSS predictors. The hazard ratios (HR) and their 95% confidence intervals (CI) were calculated. Kaplan–Meier survival curves were plotted for DSS, DFS, and OS, and statistically significant differences were identified using a log-rank test. To assess the predictive capability of the newly modified T categories, Harrell’s concordance index (c-index) [28] and the time-dependent Receiver Operating Characteristic (ROC) curve analysis, as explained by Heagerty et al. [29], were utilized to compute the integrated area under the curve (AUC). Differences with *p*-values less than 0.05 were deemed to be statistically significant. Statistical analyses were conducted using the Statistical Package for the Social Sciences (version 24.0) and R software (version 4.3.1).

## 3. Results

### 3.1. Baseline Characteristics and Comparisons between T Categories and T3b Subcategories According to Size Range

Table 1 shows the baseline characteristics of the study population. The average age was 46.6 years (range, 11–88), and the male-to-female ratio was 1:3.8. According to the traditional T category, 5535 patients (88.1%) were classified as T1, 339 (5.4%) as T2, 59 (0.9%) as T3a, and 349 (5.6%) as T3b. Based on the T3b subcategories, the 349 patients classified as T3b were further subdivided into 239 (3.8%) as T3b-1, 90 (1.4%) as T3b-2, and 20 (0.3%) as T3b-3. The average duration of the follow-up investigation was 119.2 ± 31.7 months (range, 1–226 months).

Appendix A illustrates the comparison of baseline characteristics between T categories and T3b subcategories within the same size range (T1 vs. T3b-1, T2 vs. T3b-2, and T3a vs. T3b-3). The gETE to the strap muscles had a significant impact on DSM only in cases with tumors > 2 cm (Appendix A). In tumors ≤ 2 cm, T3b-1 tumors showed no significant difference compared to T1 in overall mortality (OM) and DSM. Within the size range of 2–4 cm, T3b-2 tumors showed an increase in disease severity markers such as multifocality, lymphatic invasion, and BRAF^V600E^ positivity compared to T2 tumors. Furthermore, in T3b-2, more advanced N categories and TNM stages were recorded. Notably, in T3b-2, there was a significant increase in both OM (11.1% vs. 3.5%; *p* = 0.012) and DSM (6.7% vs. 0.6%; *p* = 0.001) compared to T2. For tumors larger than 4 cm, the severity of the disease in T3b-3 tumors increased compared to T3a. The rates of both OM (30.0% vs. 6.8%; *p* = 0.014) and DSM (25.0% vs. 1.7%; *p* = 0.003) were significantly higher in T3b-3 compared to T3a.

### 3.2. Univariate and Multivariate Analyses for Disease-Specific Mortality Risk Factors

As indicated in Table 2, for tumors ≤ 2 cm (T1 or T3b-1), the gETE to the strap muscles did not have a significant impact on DSM (HR, 2.754; CI, 0.344–22.036; *p* = 0.340). The univariate analysis identified age and tumor size as significant risk factors. However, in the multivariate analysis, only age was identified as an independent risk factor (HR, 1.165; CI, 1.086–1.250; *p* < 0.001).

As indicated in Table 3, regarding tumors in a size range of 2–4 cm (T2 or T3b-2), age, vascular invasion, and T category emerged as significant risk factors for DSM in the univariate analysis. The multivariate analysis reaffirmed the significance of age (HR, 1.088; CI, 1.026–1.154; *p* = 0.005), vascular invasion (HR, 15.159; CI, 3.511–65.450; *p* < 0.001), and T category (HR, 11.173; CI, 2.120–58.867; *p* = 0.004), emphasizing that the DSM risk for T3b-2 is higher than that for T2.

As detailed in Table 4, for tumors larger than 4 cm (T3a or T3b-3), age, tumor size, and T category were identified as significant risk factors for DSM in the univariate analysis. In line with this, the multivariate analysis indicated that age (HR, 1.069; CI, 1.008–1.134; *p* = 0.027), tumor size (HR, 2.131; CI, 1.253–3.626; *p* = 0.005), and T category (HR, 28.902; CI, 1.984–421.006; *p* = 0.014) maintained their significance. Results in Table 3 and Table 4 underscored that gETE into the strap muscle significantly increased the DSM risk in tumors larger than 2 cm.

### 3.3. Revision of T Category Based on Survival Analysis of T3b Subcategories

In Figure 2, we conduct an analysis of DSS, DFS, and OS based on the categories T1, T2, T3a, T3b-1, T3b-2, and T3b-3. In the DSS curve analysis, T3b-1, which is characterized by a smaller primary tumor size, demonstrated a higher DSS compared to T2 (log-rank *p* < 0.001) (Figure 2a). In the survival curve analysis for DFS and OS, T3b-1 consistently exhibited a higher DSS than T2 (log-rank *p* < 0.001) (Figure 2b,c). As a result, in all aspects of DSS, DFS, and OS, the survival curves were arranged in the sequence of T1, T3b-1, T2, T3a, T3b-2, and T3b-3.

Based on the results shown in Table 2, Table 3 and Table 4 and Figure 2, tumors measuring 2 cm or less were classified as ‘T1’, regardless of the presence or absence of strap muscle invasion (Figure 3). Only for tumors larger than 2 cm, those infiltrating the strap muscles have been newly defined as T3b′ in the proposed modified category. DSS curves were plotted for both the classic T categories (T1, T2, T3a, and T3b) and the modified T categories (T1′, T2′, T3a′, and T3b′). Both staging systems stratifying DSS in the sequence of T1, T2, T3a, and T3b (T1′, T2′, T3a′, and T3b′). However, a significant difference in DSS between T3a′ and T3b-3′ was observed only in the modified T categories (log-rank *p* = 0.048).

### 3.4. Predictive Performance of Classic vs. Modified T Categories for DSS

Table 5 presents the Harrell’s c-index and the AUC of the time-dependent ROC, used to compare the predictive capabilities of the classic T category and the modified T category for DSS. Harrell’s c-index was higher in the modified T category than in the classic T category (0.8961 vs. 0.8959). The AUC of the time-dependent ROC for 5-year DSS was also higher in the modified T category compared to the classic T category (AUC, 0.8573 vs. 0.8518).

## 4. Discussion

This study demonstrates the clinical significance of tumor size in the T3b category of DTC. In cases where the tumor is ≤2 cm, there was no significant difference between DSM and OM, regardless of the presence or absence of gETE. In multivariate analyses, it was emphasized that the T3b category significantly impacts the DSM risk only in tumors larger than 2 cm, underscoring the importance of size in determining the prognosis in the T3b category. In the survival curves, including DSS, DFS, and OS, they were consistently ranked in the following order: T1, T3b-1, T2, T3a, T3b-2, and T3b-3. In summary, the modified T category, which incorporated T3b-1 into the new T1′ category, demonstrated superior predictive capability for DSS compared to the existing T category. These results highlight a significant interaction between the tumor size and gETE to the strap muscles.

DTC, primarily characterized by its favorable prognosis, has been subjected to various classifications over the years, especially concerning the definitions of the T categories [6,12,13,14,30,31]. This debate primarily focuses on the significance of ETE into the strap muscles for staging, especially after its 2017 redefinition. Numerous studies have shown that mETE does not affect the recurrence or mortality in DTC [32,33,34,35,36,37,38]. Therefore, mETE was eliminated from T3, and gETE, which only invades the strap muscle, was classified as T3b. This led to the complete exclusion of the mETE concept in staging [6,8,12,13,14,39]. Despite this, the debate regarding the prognostic impact of gETE continues unabated. Some studies indicate that gETE is a significant risk factor for recurrence and mortality [24,25,26], while other studies suggest that gETE does not impact the prognosis [16,18,19,20,21].

Song et al. reported that there was no significant difference in DSS when comparing cases with gETE to those without gETE [19,21]. The reason is presumed to be that the majority of T3b tumors are under 2 cm in size. In our study, out of 349 patients diagnosed with T3b, 239 patients were classified under T3b-1 with tumors less than 2 cm, accounting for 68.5%. In the classic T category, there was no difference in the DSS between T3a and T3b, which supports the findings of previous studies. However, in the modified T category, when T3b-1 was downstaged to T1, a significant difference was observed in the DSS between T3a′ and T3b′. This reemphasizes that in order to accurately reflect the impact of strap muscle invasion, it is necessary to concurrently evaluate tumor size.

Numerous studies have compared mETE and gETE in patients diagnosed with DTC [35,40,41,42,43]. The fact that mETE and gETE present different prognoses can be extrapolated to explain that the influence of ETE changes according to tumor size. The majority of these studies indicate that the presence of gETE usually suggests a worse prognosis than mETE, including higher rates of recurrence or mortality [35,41,42]. According to the research conducted by Park et al., mETE has a more favorable prognosis than gETE, and the extent of ETE impacts the recurrence of the tumor [43]. This distinction is based on whether it can be seen under a microscope or with the naked eye, but there is a need for a more objective and accurate size standard. According to our study, the criterion is determined to be 2 cm based on the objective results of comprehensive analyses.

Another possible explanation for the varying impact of T3b depending on the tumor size could be that the possibility of complete surgical removal of the tumor may differ based on tumor size. In the MACIS (Metastases, Age, Completeness of resection, Invasion, Size) scoring system, a key indicator for assessing the prognosis of thyroid cancer, the principle of complete resection has been deemed to be significant for a long time [44,45]. However, due to the lack of standardized guidelines regarding the extent of strap muscle resection, many surgeons rely more on visual assessment than on confirming the safe margin through frozen-section analysis, even when they encounter a T3b stage intraoperatively [46]. Khan et al. reported in a study of the National Cancer Database, which involved a large cohort of 14,471 individuals, that the size of a large tumor significantly influenced margin positivity (*p* = 0.021), and margin positivity significantly decreased survival rates (*p* = 0.038) [47,48]. If tumor size is less than 2 cm, there is a possibility that the depth of invasion into the strap muscle may also be less, potentially making R0 resection more achievable. Shaha also emphasized the importance of complete resection in gETE [49]. Based on the aforementioned studies, the critical prognostic determinant within gETE may be the completeness of the tumor excision, which is affected by the tumor size.

To the best of our knowledge, there has been no study so far that has subdivided T3b based on tumor size using the same criteria as applied to T1, T2, and T3a. Furthermore, we compared the performance of the new stage with that of the traditional stage. In numerous prior stage comparison studies, Harrell’s c-index has been employed, functioning as an objective measure to evaluate the predictive ability of a model [9,28,50,51]. Furthermore, by calculating the AUC of the time-dependent ROC [29,52], it was demonstrated that the modified T category in this study is superior in evaluating DSS using both evaluation methods.

This study has clear strengths. First, it is emphasized by a comprehensive cohort that includes 6282 patients with DTC, who have been tracked for nearly a decade, providing substantial long-term follow-up. Second, the comprehensive analysis evaluated not only DFS, but also DSS and OS, providing a holistic perspective on prognosis, which includes not only recurrence, but also DSM. It is particularly noteworthy that this study explored the DSM, a crucial indicator ideally suited for evaluating the AJCC-TNM staging system. Finally, a notable characteristic of our research is that it provides objective indicators through the prediction of stage performance using various methodologies, including Harrell’s c-index and time-dependent ROC. This has enabled a comparative analysis between the existing staging and the modified staging, further strengthening the robustness of our research results.

Nonetheless, this study has several limitations. First, this is a retrospective study from a single center, which may carry the potential for selection bias. Second, the evaluation of gETE relied on the surgeon’s visual judgment, introducing a subjective element that could be susceptible to observation bias. Developing a method to evaluate ETE with standardized procedures could provide substantial benefits in the future.

## 5. Conclusions

In conclusion, T3b with a smaller tumor size (≤2 cm) demonstrated no significant difference in DSS and DFS compared to T1 in the current 8th edition of the AJCC-TNM Staging system. The modified T category, which reclassifies T3b (≤2 cm) as T1, demonstrated a more efficient performance than the existing category. If the smaller T3b is reclassified to T1, the new stage could potentially indicate a better stratification for prognosis, possibly eliminating the need for aggressive treatment.

## Figures and Tables

**Figure 1 cancers-16-02577-f001:**
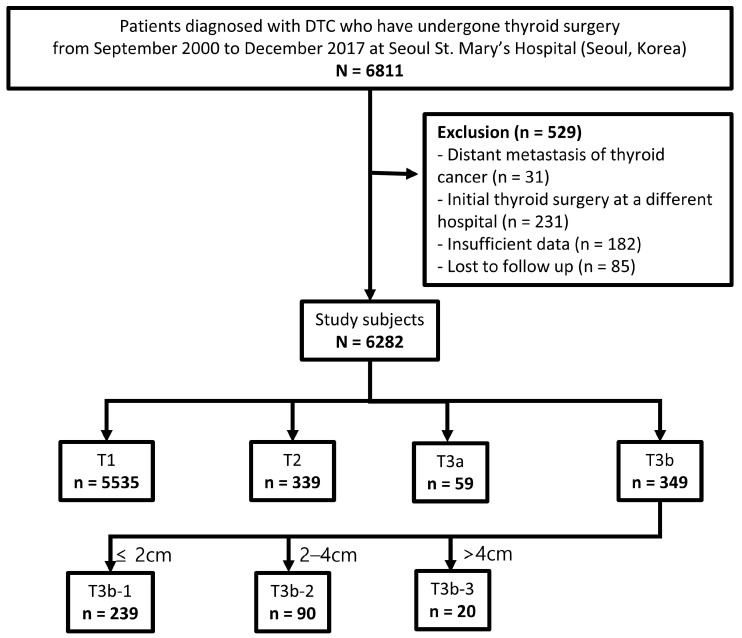
Participant flow diagram of patient selection and T3b subcategories.

**Figure 2 cancers-16-02577-f002:**
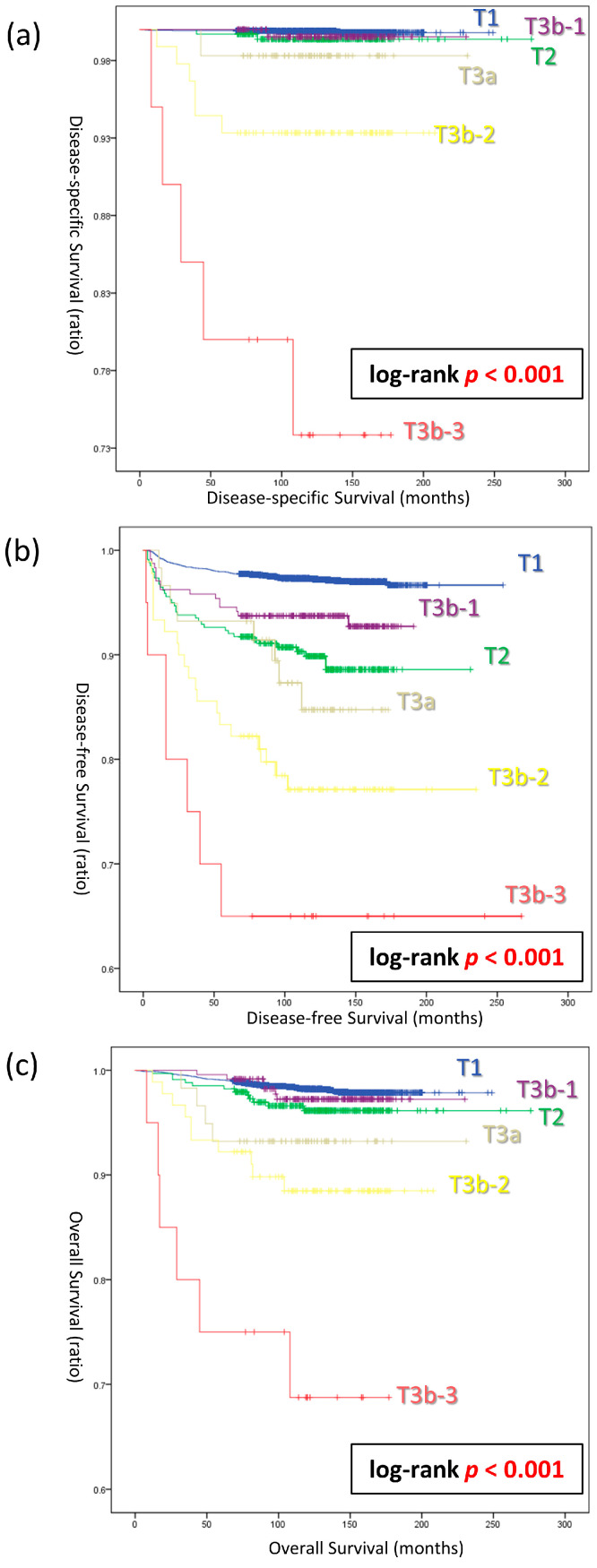
Survival analysis based on T3b subcategories. (**a**) Disease-specific survival curves (log-rank *p* < 0.001). (**b**) Disease-free survival curves (log-rank *p* < 0.001). (**c**) Overall survival curves (log-rank *p* < 0.001).

**Figure 3 cancers-16-02577-f003:**
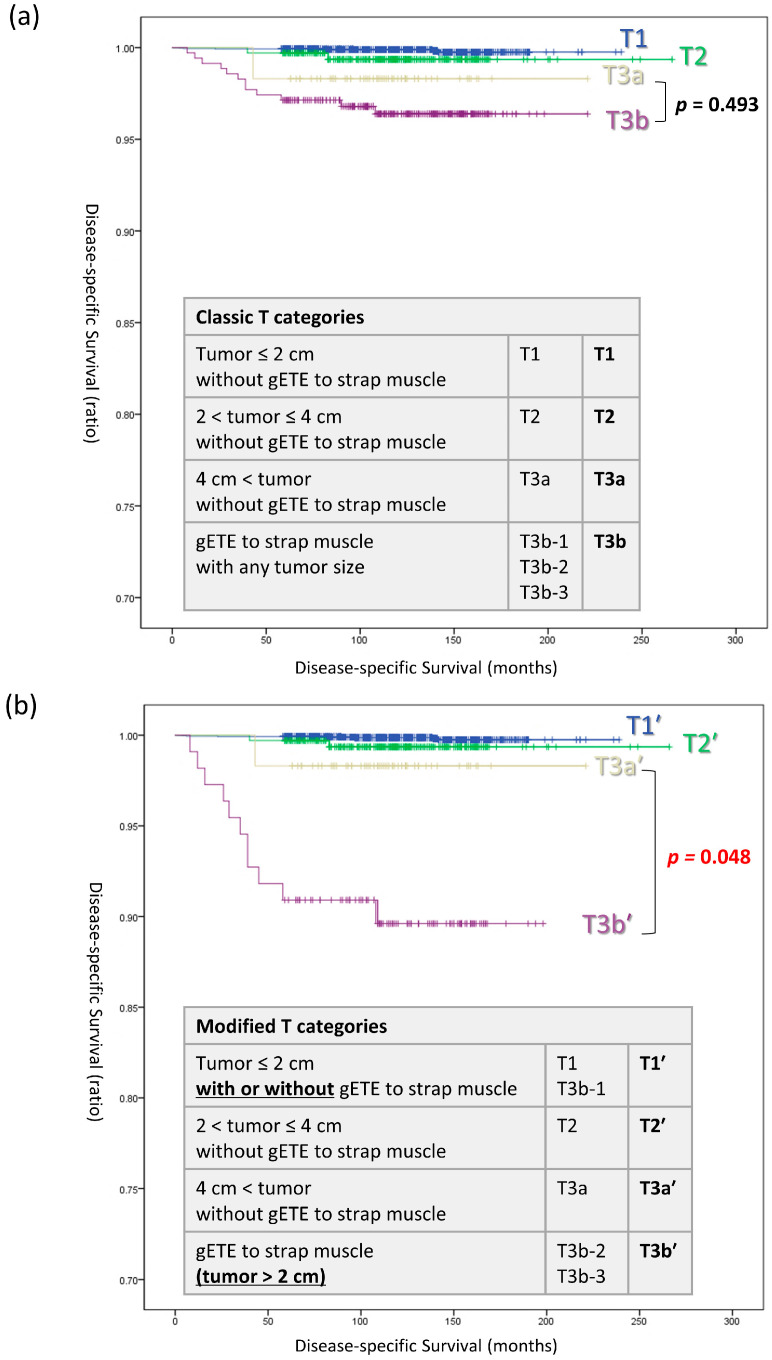
Disease-specific survival curves in (**a**) classic T categories and (**b**) modified T categories. Abbreviations: T, tumor; gETE, gross extrathyroidal extension.

**Table 1 cancers-16-02577-t001:** Baseline clinicopathological characteristics of the study population.

Total 6282 Patients	
Age (years)	46.6 ± 12.3 (range, 11–88)
Male:Female	1:3.8
Male	1311 (20.9%)
Female	4971 (79.1%)
Extent of surgery	
Lobectomy	4054 (64.5%)
TT and/or mRND	2228 (35.5%)
PTC variant	
Non aggressive	5889/6177 (95.3%)
Aggressive	288/6177 (4.7%)
FTC type	
Minimally invasive	76/80 (95.0%)
Widely invasive	4/80 (5.0%)
Tumor size (cm)	1.0 ± 0.8 (range, 0.01–13.50)
Multifocality	2306/6281 (36.3%)
Lymphatic invasion	1661/5898 (28.2%)
Vascular invasion	176/5843 (3.0%)
Perineural invasion	148/5845 (2.5%)
BRAF^V600E^ positivity	4101/5118 (80.1%)
Harvested LNs	12.9 ± 16.0 (range, 0–185)
Positive LNs	2.3 ± 4.7 (range, 0–74)
T stage	
T1	5535 (88.1%)
T2	339 (5.4%)
T3a	59 (0.9%)
T3b	349 (5.6%)
T3b-1/T3b-2/T3b-3	239 (3.8%)/90 (1.4%)/20 (0.3%)
N stage	
N0 + Nx/N1a/N1b	3392 (54.0%)/2309 (36.8%)/581 (9.2%)
TNM stage	
Stage I/II	5566 (88.6%)/716 (11.4%)
Recurrence	242 (3.9%)
Overall mortality	139 (2.2%)
Disease specific mortality	23 (0.4%)
Follow-up duration (months)	119.2 ± 31.7 (range, 1–266)

Data were expressed as number (%) or mean ± standard deviation. Abbreviations: TT, total thyroidectomy; mRND, modified radical neck dissection; PTC, papillary thyroid carcinoma; FTC, follicular thyroid carcinoma; LN, lymph node; T, tumor; N, node; M, metastasis.

**Table 2 cancers-16-02577-t002:** Univariate and multivariate analyses of disease-specific mortality risk factors in patients with T1 and T3b-1 (≤2 cm).

	Univariate		Multivariate	
	HR (95% CI)	*p*-Value	HR (95% CI)	*p*-Value
Age	1.178 (1.096–1.266)	<0.001	1.165 (1.086–1.250)	<0.001
Gender				
Female	ref.			
Male	1.236 (0.257–5.955)	0.792		
Extent of surgery				
Lobectomy	ref.			
TT and/or mRND	0.295 (0.036–2.396)	0.253		
Aggressiveness (Aggressive variant PTC, Widely invasive FTC)	0.047 (0.000–1,894,101.517)	0.732		
Tumor size	3.895 (1.013–14.980)	0.048	2.567 (0.661–9.972)	0.173
Multifocality	0.5241 (0.109–2.525)	0.420		
Lymphatic invasion	2.322 (0.623–8.654)	0.209		
Vascular invasion	6.420 (0.803–51.354)	0.080		
BRAF^V600E^ positivity	0.742 (0.077–7.134)	0.796		
T category				
T1	ref.			
T3b-1	2.754 (0.344–22.036)	0.340		
N category				
N0 + Nx	ref.	0.919		
N1a	0.939 (0.224–3.932)	0.931		
N1b	1.494 (0.175–12.791)	0.714		

Data are expressed as hazard ratio (HR) and 95% confidence interval (CI). A *p*-value < 0.05 was considered to be statistically significant. Abbreviations: TT, total thyroidectomy; mRND, modified radical neck dissection; PTC, papillary thyroid carcinoma; FTC, follicular thyroid carcinoma; T, tumor; N, node.

**Table 3 cancers-16-02577-t003:** Univariate and multivariate analyses of disease-specific mortality risk factors in patients with T2 and T3b-2 (2–4 cm).

	Univariate		Multivariate	
	HR (95% CI)	*p*-Value	HR (95% CI)	*p*-Value
Age	1.089 (1.033–1.148)	0.001	1.088 (1.026–1.154)	0.005
Gender				
Female	ref.			
Male	1.965 (0.491–7.857)	0.339		
Extent of surgery				
Lobectomy	ref.			
TT and/or mRND	0.035 (0.000–63.796)	0.382		
Aggressiveness (Aggressive variant PTC, Widely invasive FTC)	0.043 (0.000–27,053.951)	0.644		
Tumor size	2.263 (0.692–7.397)	0.177		
Multifocality	4.556 (0.919–22.572)	0.063		
Lymphatic invasion	3.280 (0.662–16.249)	0.146		
Vascular invasion	8.236 (2.059–32.946)	0.003	15.159 (3.511–65.450)	<0.001
BRAF^V600E^ positivity	0.448 (0.028–7.158)	0.570		
T category				
T2	ref.			
T3b-2	11.633 (2.348–57.638)	0.003	11.173 (2.120–58.867)	0.004
N category				
N0 + Nx	ref.	0.451		
N1a	0.000 (0.000–3.772^282^)	0.969		
N1b	2.512 (0.600–10.511)	0.207		

Data are expressed as hazard ratio (HR) and 95% confidence interval (CI). A *p*-value < 0.05 was considered to be statistically significant. Abbreviations: TT, total thyroidectomy; mRND, modified radical neck dissection; PTC, papillary thyroid carcinoma; FTC, follicular thyroid carcinoma; T, tumor; N, node.

**Table 4 cancers-16-02577-t004:** Univariate and multivariate analyses of disease-specific mortality risk factors in patients with T3a and T3b-3 (>4 cm).

	Univariate		Multivariate	
	HR (95% CI)	*p*-Value	HR (95% CI)	*p*-Value
Age	1.103 (1.029–1.182)	0.006	1.069 (1.008–1.134)	0.027
Gender				
Female	ref.			
Male	1.671 (0.337–8.295)	0.530		
Extent of surgery				
Lobectomy	ref.			
TT and/or mRND	0.028 (0.000–39.581)	0.333		
Aggressiveness (Aggressive variant PTC, Widely invasive FTC)	0.040 (0.000–72,933.798)	0.661		
Tumor size	1.464 (1.138–1.884)	0.003	2.131 (1.253–3.626)	0.005
Multifocality	0.288 (0.034–2.469)	0.256		
Lymphatic invasion	4.307 (0.479–38.720)	0.192		
Vascular invasion	0.028 (0.000–84.877)	0.382		
T category				
T3a	ref.		ref.	
T3b-3	15.860 (1.851–135.859)	0.012	28.902 (1.984–421.006)	0.014
N category				
N0 + Nx	ref.	0.868		
N1a	99,033.360 (0.000–1.084^168^)	0.952		
N1b	176,889.897 (0.000–1.929^168^)	0.950		

Data are expressed as hazard ratio (HR) and 95% confidence interval (CI). A *p* value < 0.05 was considered to be statistically significant. Abbreviations: TT, total thyroidectomy; mRND, modified radical neck dissection; PTC, papillary thyroid carcinoma; FTC, follicular thyroid carcinoma; T, tumor; N, node.

**Table 5 cancers-16-02577-t005:** Comparison of Staging systems in Classic T categories and Modified T categories.

		Harrell’s c-Index	AUC in Time-Dependent ROC for 5-Year DSS
Classic T categories	n	0.8959	0.8518
T1	5535		
T2	339		
T3a	59		
T3b	349		
Modified T categories	n	0.8961	0.8573
T1′	5774		
T2′	339		
T3a′	59		
T3b′	110		

Abbreviations: T, tumor; n, patient number; c-index, concordance index; AUC, area under the curve; ROC, Receiver Operating Characteristic; DSS, disease-specific survival.

## Data Availability

The data that support the findings of this study are available on request from the corresponding author. The data are not publicly available due to privacy or ethical restrictions.

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
