# Peer review of "Impact of Tumor Size on Prognosis in Differentiated Thyroid Cancer with Gross Extrathyroidal Extension to Strap Muscles: Redefining T3b"

_cancers, 2024, doi:10.3390/cancers16142577_

Round 1

Reviewer 1 Report

Comments and Suggestions for Authors

            COMMENTS  

The manuscript titled “Impact of Tumor Size on Prognosis in Differentiated Thyroid Cancer with Gross Extrathyroidal Extension to Strap Muscles: Redefining T3b” of Park J et al., reports an investigation about the impact of tumor size on T3b differentiated thyroid cancer (DTC) prognosis.

The aim of this study was to examine the significance of tumor size in T3b category of TNM staging system through analysis of disease-specific mortality (DSM) and disease-specific survival (DSS).

This was done because of prognostic significance of tumor size in T3b DTC still remains debated and underexplored. In fact, there are not clear indications about the stratification of DSS corresponding to each TNM stage of DTC.

By retrospective cohort analysis, this monocentric investigation has reviewed 6282 patients affected by DTC who underwent thyroid surgery at Seoul St. Mary’s Hospital from September 2000 to December 2017. Ultimately, a total of 6282 patients were investigated.

DTC T3b were subclassified into three subcategories: T3b-1 (≤2 cm), T3b-2 (2–4 cm), and T3b-3 (>4cm).

To perform this study, statistical assays were employed.

The results of this study can be grouped in three points:

1.      No significant difference was found in the prognosis of small T3b tumors compared to the T1 tumors.

2.      DSS, disease-free survival, and overall survival were significantly lower only in large T3b tumors compared to T2 and T3a. Further, if T3b tumors are 2 cm or smaller, downstaging may be considered.

3.      The modified T category, reclassifying T3b (≤2 cm) as T1, showed better staging performance than the existing category.  

In conclusion,

By adopting this modified T category, the prognostic accuracy of the AJCC/TNM staging could improve.

Simple Summary:

The “simple summary” section is adequately summarizing this study.

Abstract:

The “abstract” section is adequately describing this study.

Introduction:

This section is adequately describing the aims of study.

Minor:

Line 57: we should be uppercase.

Materials and Methods:     

This section provides sufficiently clear information.

Results:

This section provides detailed information.

Discussion:

The comments of discussion are appropriate for this investigation.

Conclusions:

By fitting with results, the conclusions are relevant.

Tables, Figures and Supplementary material: give a helpful visual representation of study.

References:

references are adequate.

Decision:

this manuscript is well written and interesting and surely, may be accepted after really minor revisions.

Author Response

Simple Summary:

The “simple summary” section is adequately summarizing this study.

Abstract:

The “abstract” section is adequately describing this study.

Introduction:

This section is adequately describing the aims of study.

Response 1: Thank you for your positive feedback.

Minor:

Line 57: we should be uppercase.

Response 2: We have made the edits in capital letter as per your advice. Thank you.

Materials and Methods:    

This section provides sufficiently clear information.

Results:

This section provides detailed information.

Discussion:

The comments of discussion are appropriate for this investigation.

Conclusions:

By fitting with results, the conclusions are relevant.

Tables, Figures and Supplementary material: give a helpful visual representation of study.

References:

references are adequate.

Decision:

this manuscript is well written and interesting and surely, may be accepted after really minor revisions.

Response 3: Thank you very much for the insightful comments.

We believe that our manuscript has been improved as a direct result of the review process. We hope that the revised manuscript is now suitable for publication in CANCERS.

Sincerely,

Joonseon Park, MD
Ja Seong Bae, MD, PhD

Reviewer 2 Report

Comments and Suggestions for Authors

The paper was written clearly and concisely. However, it has the significant bias of the surgeon's visual judgment regarding muscle involvement. 

Have you considered also using the analysis of the surgical specimen?

Have other mutations besides BRAFV600 been considered?

How many patients with T3b1 who underwent a lobectomy needed another surgery or RAI?

And of these, how many had an aggressive variant?

Comments on the Quality of English Language

The quality of english is excellent

Author Response

Point-by-point response to Comments and Suggestions for Authors

The paper was written clearly and concisely. However, it has the significant bias of the surgeon's visual judgment regarding muscle involvement.

Have you considered also using the analysis of the surgical specimen?

Response 1: Thank you for pointing this out. We agree with this comment. Although we did not analyze the surgical specimens, a previous smaller study conducted by our team at the same institution included photographs of T3b specimens, which I have referenced and added to the manuscript. Thank you.

Reference: Park, Joonseon, et al. "Clinical Significance of Tumor Size in Gross Extrathyroidal Extension to Strap Muscles (T3b) in Papillary Thyroid Carcinoma: Comparison with T2." Cancers 14.19 (2022): 4615.

Have other mutations besides BRAFV600 been considered?

Response 2: Thank you for your insightful feedback. At our institution, in addition to BRAF, we occasionally perform additional tests such as NRAS, TERT, among others. However, due to cost considerations for patients, these tests are not conducted routinely. Furthermore, TERT testing has been conducted since 2019 and couldn’t be included in this cohort. Consequently, we have included BRAF testing, which has been performed on a larger proportion of patients, to represent the cohort for analysis. We will include a wider range of mutation results for further analysis. Thank you.

How many patients with T3b1 who underwent a lobectomy needed another surgery or RAI?

And of these, how many had an aggressive variant?

Response 3: Thanks to you, we have re-evaluated the data. There were 24 cases of T3b-1 patients who underwent lobectomy. Among them, one patient underwent additional surgery and received RAI. Additionally, there were 3 cases of aggressive variants among the 24 cases.

4. Response to Comments on the Quality of English Language

Point 1: The quality of english is excellent

Response : Thank you for your positive feedback. I also uploaded the certificate of English correction by a native speaker through a pain service as non-published materials.

 We believe that our manuscript has been improved as a direct result of the review process. We hope that the revised manuscript is now suitable for publication in CANCERS.

Sincerely,

Joonseon Park, MD
Ja Seong Bae, MD, PhD

Reviewer 3 Report

Comments and Suggestions for Authors

Dear Dr Ja Seong Bae

It is a very interesting paper regarding an intriguing issue. As you have already mentioned the evaluation of gETE relied on the surgeon’s visual judgment and this is the major limitation when such a study is designed. Developing a method evaluating ETE with more standardized procedures could be of value. This could be a real challenge for thyroid surgeons in the future. 

Author Response

Thank you very much for taking the time to review this manuscript, which helped us improve the manuscript significantly. We are pleased to have had the chance to revise our manuscript (cancers-3050018) submitted for publication in CANCERS. Please find the detailed responses below and the corresponding revisions in track changes in the re-submitted files.

Comments 1: Dear Dr Ja Seong Bae

It is a very interesting paper regarding an intriguing issue. As you have already mentioned the evaluation of gETE relied on the surgeon’s visual judgment and this is the major limitation when such a study is designed. Developing a method evaluating ETE with more standardized procedures could be of value. This could be a real challenge for thyroid surgeons in the future.

We thank you and the reviewers for the insightful comments. We believe that our manuscript has been improved as a direct result of the review process. We hope that the revised manuscript is now suitable for publication in CANCERS.

Sincerely,

Joonseon Park, MD
Ja Seong Bae, MD, PhD
